# Antibody–Drug Conjugates in the Treatment of Genitourinary Cancers: An Updated Review of Data

Prathana Nathan [1], Adnan Rajeh [2], Meh Noor [1], Gabriel Boldt [3] and Ricardo Fernandes [2,4,*]

1  Department of Internal Medicine, Schulich School of Medicine & Dentistry, Western University, London, ON N6A 5C1, Canada
2  Division of Medical Oncology, Department of Oncology, Schulich School of Medicine & Dentistry, Western University, London, ON N6A 5C1, Canada; adnan.rajeh@lhsc.on.ca
3  London Regional Cancer Program, Victoria Hospital, London Health Sciences Centre, London, ON N6A 5W9, Canada; gabriel.boldt@lhsc.on.ca
4  Cancer Research Laboratory Program, Lawson Health Research Institute, London, ON N6C 2R5, Canada
*  Correspondence: ricardo.fernandes@lhsc.on.ca

**Abstract:** The treatment landscape of genitourinary cancers has significantly evolved over the past few years. Renal cell carcinoma, bladder cancer, and prostate cancer are the most common genitourinary malignancies. Recent advancements have produced new targeted therapies, particularly antibody–drug conjugates (ADCs), due to a better understanding of the underlying oncogenic factors and molecular mechanisms involved. ADCs function as a 'drug delivery into the tumor' system. They are composed of an antigen-directed antibody linked to a cytotoxic drug that releases cytotoxic components after binding to the tumor cell's surface antigen. ADCs have been proven to be extremely promising in the treatment of several cancer types. For GU cancers, this novel treatment has only benefited patients with metastatic urothelial cancer (mUC). The rest of the GU cancer paradigm does not have any FDA-approved ADC treatment options available yet. In this study, we have thoroughly completed a narrative review of the current literature and summarized preclinical studies and clinical trials that evaluated the utility, activity, and toxicity of ADCs in GU cancers, the prospects of ADC development, and the ongoing clinical trials. Prospective clinical trials, retrospective studies, case reports, and scoping reviews were included.

**Keywords:** genitourinary cancers; bladder carcinoma; urothelial carcinoma; kidney cancer; renal cell carcinoma; antibody–drug conjugates

## 1. Introduction

Genitourinary (GU) malignancies are represented by a heterogenous group of cancers with various histology and biology, including renal cell carcinoma, bladder cancer, and prostate cancer. The treatment strategies for genitourinary malignancies have significantly evolved over the last two decades [1]. More recently, antibody–drug conjugates (ADCs) have gained attention in many different cancers. For instance, in the GU realm, a remarkable positive outcome has been shown with the combination of the antibody–drug conjugate (enfortumab vedotin) with pembrolizumab for the treatment of metastatic urothelial carcinoma [2].

ADCs have a unique characteristic and include a monoclonal antibody (mAb) bound to a cytotoxic molecule (payload), with a linker in between. These novel drugs are called "smart chemotherapy delivery" due to their ability to bind to the target (antigen on the surface of the cancer cell) and subsequently release their cytotoxic payload into the cancer cell [3]. Once the ADCs enter the cancer cell, they are degraded in the lysosomes [4], releasing the active payloads, causing apoptosis and antibody-dependent cellular cytotoxicity (ADCC) [5]. In addition to that direct effect, cellular damage to the adjacent cancer cells occurs due to the diffusion of the payload into the tumor microenvironment, which is known as the "bystander effect". The Fab portion also has the ability to stimulate the

immune system cells, activating an immune-mediated response that has the potential to incur further harm to the cancer cells [6,7].

The structure of the ADC is extremely important to understand in order to appreciate its mechanism of action. It consists of a monoclonal antibody, a linker, and a payload. The monoclonal antibody must be designed properly in order to exhibit its therapeutic abilities and reduce off-target toxicity [8]. The antigen on the cancer cell should be present on the cell surface to allow for the binding of the antibody and the ability for internalization of the ADC into the cell [9–11]. The linker supports molecule stability and payload release, and ultimately leads to the timely release of the payload at the target cell [12]. There are two types of linker molecules: cleavable and non-cleavable [13]. Some characteristics of a payload are key for them to work well against cancer, which includes conjugation accessibility and drug potency [14]. In addition, anti-mitotic agents (microtubule inhibitors and topoisomerase inhibitors) or DNA damaging toxins are also part of the ADCs' machinery [15].

To date, twelve ADCs have been approved by the FDA and nine by the EMA [15–17]. Only four of these ADCs have uses in genitourinary cancers, most of them for metastatic urothelial carcinoma. Interestingly, there are data for ADCs in patients with non-muscle-invasive urothelial carcinoma (NMIBC). A study suggests promising activity of ADCS in patients with BCG-refractory NMIBC disease [18]. There are six ADCs in total that are approved or being investigated for genitourinary cancers, and this narrative review will focus on the following three of them: enfortumab vedotin (EV), sacituzumab govitecan (SG), and trastuzumab deruxtecan (TD).

## 2. Methods

For this narrative review, a literature search was conducted on PubMed, Cochrane, and Embase databases, using the terms "genitourinary cancers", "prostate cancer" "bladder carcinoma", "urothelial carcinoma", "kidney cancer", and "renal cell carcinoma", along with any of the following: "antibody–drug conjugates" (ADCs), "enfortumab vedotin" (EV), "sacituzumab govitecan" (SG), and "trastuzumab deruxtecan", from January 2003 to December 2023. The same search terms were used for the ClinicalTrials.gov registry of clinical trials. Abstracts from the annual meetings for the American Society of Clinical Oncology (ASCO) and the European Society for Medical Oncology (ESMO) were included. Only English studies were included.

All prospective clinical trials, either comparative or noncomparative, were included. Retrospective studies, case reports, and literature reviews were also included if they had a specific angle that added value to the paper's objective. Only English studies were included.

## 3. Results

After removing any duplicates, two independent reviewers selected the articles. Roughly 225 papers were screened, and ultimately 58 studies were selected for review. The reviewers extracted the necessary information from the chosen studies and incorporated the data into the study's paragraphs and tables as a narrative review (see Figure 1).

### 3.1. ADCs in the Treatment of GU Cancers: Time for Loaded Guns

Currently, ADCs have not been part of the treatment landscape of metastatic renal cell carcinoma. A phase I trial enrolling patients with different cancers, including RCC, has evaluated vobramitamab duocarmazine (MCG018), an ADC that targets B7-H3 (CD276) [19].

With regards to metastatic prostate cancer, trials that have assessed the efficacy of ADCs, such as sacituzumab govitecan, did not show any meaningful clinical benefit. However, ongoing trials investigating the clinical role of novel compounds targeting prostate cancer antigens have not been reported yet. These studies are evaluating the role of ARX517, which targets PSMA, vobramitamab duocarmazine (MGC018), and AMG 509, which targets STEAP1. The available results highlight PSMA-based ADCs as promising agents [20]. In general, there are still no approved ADCs for the treatment of metastatic prostate cancer.

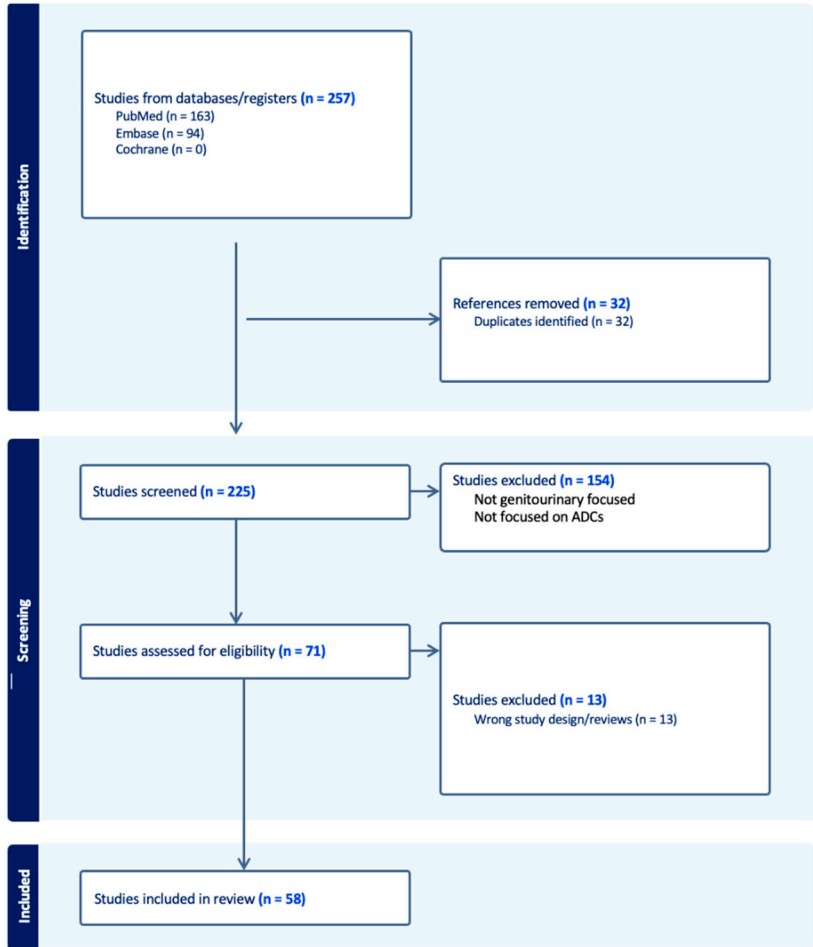

**Figure 1.** Schematic representation of the identification, screening, and inclusion process of the selected studies.

There is no evidence for the use of ADCs in any of the stages of testicular cancer, and there are no ongoing trials based on this literature review.

Most of the evidence for the use of ADCs is for metastatic urothelial carcinoma (mUC) [21]. This review includes and highlights three drugs, enfortumab vedotin, sacituzumab govitecan, and trastuzumab deruxtecan. The other ADCs that are currently being investigated for mUC do not have reported results, and there is lack of evidence to support their utility in treating mUC [22].

### 3.2. Enfortumab Vedotin

Enfortumab Vedotin (EV), in the context of treating advanced urothelial carcinoma, shows notable efficacy and safety profiles. It targets Nectin-4, a protein found on urothelial carcinoma cells, allowing for the targeted delivery of the cytotoxic agent [23–26]. The clinical trials have demonstrated significant response rates in patients with metastatic urothelial carcinoma (UC), including those who have previously received platinum-containing chemotherapy and PD-1 or PD-L1 inhibitor therapy [27–29]. In terms of safety, the common adverse effects include fatigue, a skin rash, decreased appetite, neuropathy, and dysgeusia. The balance of efficacy and manageable side effects positions EV as a promising option in the treatment landscape of metastatic UC [30–32].

Three major trials have been pivotal in the development of EV. EV-101 is a phase I trial focusing on dose escalation and expansion in patients with Nectin-4-positive tumors, including metastatic UC [33,34]. It included 155 patients in the mUC cohort. It assessed the safety and efficacy of EV. The recommended dose of EV was identified as 1.25 mg/kg. The most common treatment-related adverse events (TRAEs) identified were a skin rash,

peripheral neuropathy, fatigue, alopecia, and nausea. These TRAEs were grade 1–2 in severity. In the EV-101 trial, EV demonstrated an objective response rate (ORR) of 43%, regardless of any previous treatment, indicating its effectiveness in mUC treatment. The median overall survival (OS) was established as 12.3 months, and the OS rate at 1 year was 51.8%.

EV-201, a muti-center phase II trial, investigated EV in patients who had been previously treated with platinum-based chemotherapy and immune checkpoint inhibitors (ICI). This study showed positive outcomes and led to a subsequent phase 3 trial. EV demonstrated an ORR of 44.0% in patients who had received prior platinum-based chemotherapy and ICI, including a 12% complete response, and an ORR of 52% was seen in cisplatin-ineligible patients who had been treated initially with ICI, with 20% achieving a complete response. Overall, 17% of the patients experienced treatment-related adverse events, and the most common grade 3 or greater included neutropenia (9% of patients), maculopapular rash (8% of patients), and fatigue (7% of patients) [35].

EV-301 is a phase III trial that compared EV with chemotherapy in patients who had progressed on platinum-based chemotherapy and immune checkpoint inhibitors. While 301 patients were randomized to EV, 307 patients were randomized to receive chemotherapy. Enfortumab vedotin demonstrated a 30% lower risk of death than chemotherapy, indicating significantly longer overall survival. The median progression-free survival (mPFS) was 5.55 months in the EV group, compared to 3.71 months in the chemotherapy group, with a hazard ratio for progression or death of 0.62 (95% CI, 0.51 to 0.75; $p < 0.001$). The incidence of treatment-related adverse events was high overall but was similar in the two groups, with 93.9% in the EV group and 91.8% in the chemotherapy group experiencing these events. This study was crucial for establishing the drug's superiority over standard chemotherapy in terms of overall survival and progression-free survival [36], which led to the approval of EV in this patient population [37].

Finally, EV-302/KEYNOTE-A39 is a phase 3, open-label study, including patients with treatment-naive mUC who were randomized to EV plus pembrolizumab (P) versus platinum-based chemotherapy. This was a practice-changing trial, as it showed a significant improvement of PFS (median PFS, 12.5 months versus 6.3 months, respectively; HR 0.45; $p < 0.00001$) and OS (median OS, 31.5 months versus 16.1 months, respectively; HR 0.47; $p < 0.00001$). Furthermore, the responses rates were remarkably better for those patients treated with EV plus P (ORR 67.7%) than those treated with chemotherapy (ORR 44.4%). In terms of safety, overall, 55.9% of the patients in the EV plus P arm experienced grade 3 or greater TRAEs compared to 69.5% of those patients treated with chemotherapy. The common grade 3 and 4 side effects were a maculopapular rash (7.7%), hyperglycemia (5.0%), and neutropenia (4.8%) in the EV + P arm, and hematologic toxicities for the chemotherapy group, including anemia (31.4%), neutropenia (30.0%), and thrombocytopenia (19.4%) [2,38].

EV has been reported to be well-tolerated, with common treatment-related adverse events including fatigue, alopecia, decreased appetite, peripheral neuropathy, a skin rash, and hyperglycemia [39,40]. In the EV-301 trial, the major adverse events associated with EV in the study included skin reactions, peripheral neuropathy, and hyperglycemia [28,41,42]. The incidence of treatment-related adverse events with EV was manageable and similar to that of chemotherapy in the EV-301 trial [17,43,44].

Various case studies have illustrated other possible side effects of EV. There is a report of two cases of asthma exacerbation; however, this was noted to be in patients with a history of childhood asthma, active smoking, and atopic dermatitis [45]. This report highlights that careful attention should be paid to respiratory symptoms in the early stages of EV treatment in those with risk factors for asthma. There have been a few case reports of severe eczematoid and lichenoid eruption with full-thickness epidermal necrosis developing from metastatic UC treated with EV, including a case of a patient who developed toxic epidermal necrolysis on Day 12, ultimately resulting in death on Day 18 [46–49]. These case reports

underline the importance of vigilance for early skin- and respiratory-related adverse effects in order to decide whether to continue with subsequent doses of EV [50].

### 3.3. Trastuzumab Deruxtecan

The family of human epidermal growth factor receptors are involved in the signal transduction pathways required for cell proliferation. Human epidermal growth factor receptor-2 (HER2) proliferation is observed in 2–3% of solid tumors and is associated with more aggressive metastatic disease, making it of clinical significance. More specifically, it is overexpressed in approximately 30% of breast cancers and 10–20% of gastric malignancies. HER2 amplification in urothelial carcinomas is reported to range between 8.5 and 81%, with wide variability possibly attributed to tumor heterogeneity [51].

Trastuzumab deruxtecan (T-DXd), better recognized under its brand name Enhertu, is a targeted antibody–drug conjugate composed of a humanized monoclonal antibody covalently linked to a topoisomerase I inhibitor (DXd). The safety and efficacy of T-DXd were demonstrated in a population of patients with advanced treatment-refractory, HER2-positive malignancy breast and gastric cancers, for the most part [52]. In this first phase 1 dose-escalation study, 22 patients with HER2-positive breast cancer, gastric cancer, or other HER2-expressing solid tumors were treated with T-DXd once every three weeks, at a dose ranging from 0.8 mg/kg to 8.0 mg/kg. Notably, no dose-limiting toxicity was observed, and a maximal tolerated dose was not achieved. The target drug exposure was noted at the dose of 6.4 mg/kg, which was, subsequently, the dose for phase 2. In 2019, the Food and Drug Administration (FDA) accelerated approval for T-DXd to be used for advanced HER2-positive breast cancer. Thus far, HER2-targeted therapies are FDA-approved only for breast, gastric, and gastroesophageal cancers [53]. More recently, T-DXd in combination with nivolumab has depicted antitumor activity in a phase 1b study of patients with HER2-expressing advanced/metastatic urothelial carcinoma after previous platinum-based chemotherapy [54].

The MyPathway trial (NCT02091141) evaluated FDA-approved targeted therapies in non-indicated advanced solid tumors with relevant molecular changes. The HER2 basket consisted of patients with HER2-altered tumors treated with pertuzumab and trastuzumab. The anti-dual HER2 treatment was found to be active in a wide range of KRAS wild-type, HER2-amplified tumors; however, limited activity was seen in KRAS-mutated tumors [53]. This suggests that alterations at the molecular level may be of clinical relevance, especially in the context of tailoring treatment options.

Hussain et al. 2007 investigated the HER2 overexpression rate in advanced urothelial carcinoma patients and assessed the safety and efficacy of a regimen involving trastuzumab, carboplatin, gemcitabine, and paclitaxel. A total of 70% of the patients responded completely or partially, with median time to progression and survival times of 9.3 and 14.1 months, respectively. A total of 22.7% experienced cardiac toxicity, which was higher than projected, however, the vast majority were grade 2 or lower. The group concluded that, ultimately, a randomized controlled trial would be needed to determine the true contribution of trastuzumab [55]. A subsequent multi-center, randomized phase II trial assessed gemcitabine and platinum salt, with or without trastuzumab, in patients with locally advanced or metastatic urothelial carcinoma with HER2 overexpression. Due to the low incidence of HER2 overexpression, the findings were of limited clinical significance. In an exploratory analysis, those patients treated with trastuzumab and cisplatin, compared to carboplatin-based chemotherapy, fared better [56]. In both of the studies briefly described above, myelosuppression was the main grade 3/4 toxicity. The commonly reported dose-limiting toxicities included neutropenia and gastrointestinal toxicity [57]. Interstitial lung disease (ILD) and pneumonitis are included in the black box warning for patients treated with T-DXd.

### 3.4. Other HER2-Targeting ADCs

The antibody–drug conjugate trastuzumab emtansine (T-DM1) is currently approved for HER2-positive breast cancer treatment. The phase II KAMELEON trial looked at the relationship between HER2 expression and trastuzumab emtansine (T-DM1) in patients with biomarker-positive urothelial bladder cancer or pancreatic cancer/cholangiocarcinoma. Although the trial was terminated prematurely, due to recruitment difficulty, the study suggested that patients with HER2-positivity can respond to T-DM1, thereby potentially informing the use of T-DM1 in HER2-positive non-breast cancers [58].

Trastuzumab duocarmazine (TD) is a novel targeted ADC with several preclinical trials promising antitumor activity. A phase 1 dose-escalation and dose-expansion study in 2019 was the initial in-human study assessing its safety in advanced solid tumor treatment [59]. Clinical activity and safety were observed in heavily pretreated patients with HER2-positive metastatic cancers, including those resistant to T-DM1, as well as those with tumors with a low expression of HER2. Notably, one dose-limiting toxic effect was death from pneumonitis, which occurred at the highest administered dose in the dose-escalation phase.

### 3.5. Sacituzumab Govitecan

Sacituzumab govitecan (SG) is an antibody targeting the trophoblast cell-surface antigen (Trop-2) conjugated with a DNA-disrupting agent, SN-38, with a pH-sensitive cleavable linker [60,61]. Trop-2, a surface glycoprotein expressed on many epithelial tumors, including urothelial cancer, is a suitable target for this novel approach [62]. The ADC delivers SN-38, which is the active metabolite of irinotecan, to the tumor cells [63].

The initial phase 1 trial completed on SG looked at the treatment of a diverse set of metastatic tumors [64]. The study demonstrated promising results in patients with refractory diseases, including two patients with a partial response (triple-negative breast cancer or colon cancer) and sixteen patients with stable disease. Notably, the drug demonstrated anti-tumor activity in those patients who had previously failed or relapsed on topoisomerase-I-inhibitor-containing regimens [65,66].

Neutropenia was identified as the dose-limiting toxicity, with 12 mg/kg being the maximum tolerated dose for cycle 1. However, this dose was deemed too toxic for repeated cycles. Lower doses (8 and 10 mg/kg) were better tolerated for extended treatment, with no treatment-related grade 4 toxicities being reported. Grade 3 toxicities were limited to fatigue, neutropenia, diarrhea, and leukopenia. The other common toxicities included mild diarrhea, fatigue, nausea, vomiting, and alopecia, with most being grade 1 and 2 [65–67].

TROPHY-U-01, a multi-cohort phase 2 trial evaluating the efficacy of SG in mUC, enrolled patients previously treated with platinum-based chemotherapy and ICI. The objective response rate (ORR) was 27%, which included 6 complete responses and 25 partial responses, indicating significant anticancer activity in this patient population. The median OS was reported to be 10.9 months (95% CI, 9.0 to 13.8 months). The median DOR was 7.2 months (95% CI, 4.7 to 8.6 months), and the median PFS was 5.4 months (95% CI, 3.5 to 7.2 months). The study demonstrated that SG has notable efficacy compared to the historical controls in pretreated mUC patients who have progressed on both prior platinum-based regimens and checkpoint inhibitors [68].

Cohort 2 of the TROPHY-U-01 trial consisted of patients considered ineligible for platinum who progressed after ICI. The preliminary results have shown that, of the 38 patients included, the ORR was found to be 32%, the median DOR was 5.6 months, and the median PFS was 5.6 months. The median time to response was 1.4 months (range, 1.3–1.5), and the median OS was 13.5 months. In terms of safety, grade--$\geq$-3-treatment-related adverse events (TRAEs) were reported in 68% of patients, with the most common grade $\geq$ 3 TRAEs being neutropenia (34%), anemia (21%), leukopenia (18%), fatigue (18%), and diarrhea (16%). These TRAEs resulted in an 18% discontinuation rate [69].

There have also been positive results from cohort 3 of the TROPHY-U-01 trial, which looked at SG in combination with pembrolizumab after progression on platinum-based chemotherapy. The ORR was 41% in the 41 patients included, the CBR was 46%, the

median DOR was 11.1 months, and the median PFS was 5.3 months. The clinical benefit rate was defined as the complete response rate plus partial response rate plus stable disease ≥ 6 months. The median time to response was 1.4 months, and the median OS was 12.7 months. No new safety signals were identified. The TRAEs led to a 15% discontinuation rate. Systemic steroid and G-CSF use were both 34% [70].

The ongoing clinical trials with ADCs in GU cancer are outlined in Table 1.

**Table 1.** Selected clinical trials investigating ADCs.

| Agent | Targets of Therapy | Trial Name | Authors | Setting | Phase | Sample Size | Completion Date |
|---|---|---|---|---|---|---|---|
| Enfortumab Vedotin | Nectin-4 | EV-101 | Grivas et al. | Nectin-4-positive patients, including advanced urothelial carcinoma | 1 | 155 patients in the mUC cohort | 23 June 2014 to 25 October 2018 |
| | | EV-202 cohort 1 | Yu et al. | Patients who had previously received both platinum-based chemotherapy and a PD-1 or PD-L1 inhibitor | 2 | 125 | 8 October 2017 to 2 July 2018 |
| | | EV-202 cohort 2 | Rosenburg et al. | Patients who had received a PD-1 or PD-L1 inhibitor and were ineligible for cisplatin | 2 | 89 | 8 October 2017 to 11 February 2020 |
| | | EV-301 | Powels et al. | Patients who had progressed on platinum-based chemotherapy and immune checkpoint inhibitors | 3 | 301 patients were randomized to EV and 307 patients were randomized to chemotherapy | Date of data cutoff was 15 July 2020 |
| | | EV-302 | Powels et al. | Patients with previously untreated la/mUC who were eligible for cisplatin- or carboplatin-containing chemotherapy | 3 | 886 | |
| Sacituzumab Govitecan | Trop-2 | IMMU-132-01 | Starodub et al. | Diverse metastatic epithelial cancers | 1 | 25 | |
| | | TROPHY-U-01 Cohort 1 | Tagawa et al. | Patients with mUC after progression on platinum-based chemotherapy and ICI | 2 | 113 | 17 December 2012 to 22 June 2017 |
| | | TROPHY-U-01 Cohort 2 | Petrylak et al. | Patients considered ineligible for platinum who progressed after ICI | 2 | 28 | Study started 13 August 2018 |
| | | TROPHY-U-01 Cohort 3 | Grivas et al. | Patients with progression on platinum-based chemotherapy | 2 | 41 | Study started 13 August 2018 |
| DS-8201a (Trastuzumab Deruxtecan) | HER2 | | Hussain et al. | Trastuzumab, carboplatin, gemcitabine, and paclitaxel in advanced urothelial carcinoma patients (1) | 2 | 44 | October 2000 to March 2005 |
| | | CVH-CT0 | Oudard et al. | Gemcitabine and platinum salt, with or without trastuzumab, in patients with locally advanced or metastatic urothelial carcinoma (2) | 2 | 61 | February 2004 to October 2009 |
| | | MyPathway | Meric-Bernstam et al. | Pertuzumab (P) + trastuzumab (H) treatment of a large, tissue-agnostic cohort of patients with HER2-positive advanced solid tumors (3) | 2 | 22 | 14 April 2014 to 15 June 2020 |
| | | DS8201-A-U105 | Glasky et al. | T-DXd in combination with nivolumab in patients with HER2-expressing advanced/metastatic UC | 1b | 34 | July 2021 |

## 4. Conclusions

Antibody–drug conjugates (ADCs) have been called the "magic bullet" in cancer treatment, which is a phrase coined by Dr. Paul Ehrlich, with his concept of the ideal anticancer drug. ADCs are highly specific anticancer agents compared to chemotherapies and immunotherapies, as they have been shown to recognize, bind, and neutralize cancer cells while limiting the injury to healthy cells [71].

It has been well established at this point that ADCs have a positive effect on metastatic urothelial carcinoma in the different settings of treatment [72,73]. As for the other genitourinary cancers, there are some ongoing studies looking at different agents, but these are all largely still early phase trials. Currently, we do not have any FDA-approved ADCs for the treatment of metastatic renal cell carcinoma, testicular cancer, or prostate cancer. Despite best efforts, none of the available ADCs have been able to generate meaningful responses in any of these cancers [74].

With the recent approval of enfortumab vedotin and pembrolizumab in the first-line treatment setting for mUC and the impressive response rates associated with the Keynote-A39 study, ADCs have found their way into the treatment landscape of genitourinary cancers and are here to stay. As novel treatments, the toxicity profile of ADCs is still not well understood. It seems, though, that they are relatively safe when combined with immunotherapies, and are less toxic than the chemotherapies used to treat the same disease site, even though they usually carry payloads that are much more toxic than the standard chemotherapy drugs.

As metastatic genitourinary cancers, in general, are associated with poor survival rates, along with the wide spectrum of toxicities that conventional treatments cause, ADCs, with their highly specific ability to target cancer cells, are an attractive option for clinicians and patients alike [75,76]. This current narrative review illustrates that ADCs seem to be a rising star in cancer research and potentially the next big advancement in the cancer treatment paradigm [77,78].

**Author Contributions:** Conception and design: R.F., A.R., P.N. and G.B. Literature review: A.R., P.N., M.N. and G.B. Manuscript writing: R.F., A.R., P.N. and M.N. Final approval of manuscript: R.F., A.R., P.N. and M.N. All authors have read and agreed to the published version of the manuscript.

**Funding:** This research received no external funding.

**Institutional Review Board Statement:** The authors are accountable for all aspects of the work in ensuring that questions related to the accuracy or integrity of any part of the work are appropriately investigated and resolved.

**Conflicts of Interest:** R.F. has the following disclosures: Advisory Board or Honoraria: Bayer, Merck, Novartis, Janssen, Pfizer, BMS, Ipsen, Bayer; Travel Grant: Janssen. The authors have no other conflicts of interest to declare.

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
