# Peer review of "Antibody–Drug Conjugates in the Treatment of Genitourinary Cancers: An Updated Review of Data"

_curroncol, doi:10.3390/curroncol31040172_

Round 1
Reviewer 1 Report
Comments and Suggestions for Authors
Dear Authors,
I reviewed with interest the paper entitled “Antibody Drug Conjugates in Treatment of Genitourinary Cancers: An Updated Review of Data”.
First, I would strongly congratulate with the authors for their work for this study, which covers an interesting topic such as proving a general view of the ADCs available for GU cancers.
I found the present study interesting and well written - no major concerns with English language editing.
- The title is clear and descriptive of what authors have explored in their work – although actually is it focused on urothelial carcinoma
- The Introduction is clear, fluent to read, and provides a background which is relevant to the study, with that aim properly provided at the end of the section.
- Materials and Methods are clearly described. The paper results methodologically correct. However, with regards to the literature search, looking at the key word used, it seems that “prostate cancer” – although then mentioned, -, has not been included in the search; could you please clarify this aspect?
- Figures and Tables are clear and not repetitive, as well as the Results, which are presented in subheadings and fluent to read. With specific regards to the first Table: there is not a caption and number of the Table; moreover, the first column should be modified: it reports the full citation of the article, yet it should only mention first author, year and the proper reference linking to the reference list. Moreover, with specific regards to Table 1, the column “completion date”, just the years should be mention (day and months are not that important).
- Conclusions are well stated.
I have not further suggestions.
Author Response
Dear Reviewer,
Thanks for your valuable comments.
Please find attached the file with our responses.
We have made the changes on the updates manuscript.
Many Thanks,
Ricardo Fernandes

Reviewer 2 Report
Comments and Suggestions for Authors
In the present work Nathan et al. reviewed the literature through the summarization of preclinical studies and clinical trials that evaluated utility, activity, and toxicity of antibody drug conjugates (ADCs) in genitourinary (GU) cancers. They also reviewed the prospects of ADC development, and ongoing clinical trials. Their study also included prospective clinical trials, retrospective studies, case reports, and scoping reviews. Their work is interesting and it has merit for publication after addressing some issues.
From an organizational point of view the authors report on their search “Methods”, but then they get directly into their results, without previously having added a heading such as “Results” or “ACDs in GU” etc.
Since the authors have attempted to write a systematic review they should provide a schematic diagram with their search strategy. Meaning, initial search, articles removed, criteria, articles included etc.
The first table has no legend… it is also not necessary to provide the complete reference, a Hussain et al (year) and a citation would suffice. In addition, I did not understand what this table describes. The contents of this table are described in the text, so either drop the table or summarize the findings reported in the text in a more comprehensive way. In table 1 (which is actually table 2) the authors should provide also the citation that corresponds to the reference section of their paper.
Finally, the authors report (in the “Methods” section) that 58 studies were selected for their review, while in the “Reference” section there are 33 studies. Shouldn’t there be at least 58?
Comments on the Quality of English LanguageAuthor Response
Dear Reviewer,
Thanks for your valuable comments.
Please find attached the file with our responses.
We have made the changes on the updates manuscript.
Many Thanks,
Ricardo Fernandes

Reviewer 3 Report
Comments and Suggestions for Authors
- The aim of the paper is clear and of current interest due to the multiple recent and ongoing clinical trials of ADCs in GU malignancies. This review gives context and summarizes recent trails and is a useful review.
- The manuscript is clear relevant and presented in a well-structured manner. The methodology is appropriate.
- The manuscript is scientifically sound and the experimental design is appropriate.
- The tables summarize the relevant data appropriately. As it is a descriptive review, statistical analysis is not applicable. The conclusions are consistent with the evidence and arguments presented.
- The cited references need to be improved. The methods detail that 58 studies were selected for review however they are all not cited. The formatting for references is not uniform. Please correct that.
Author Response

(The authors gave the same response as above.)

Round 2
Reviewer 2 Report
Comments and Suggestions for Authors
The authors have addressed the previous issues I highlighted.
Remove colon (:) from the "Results" heading.
Now, the references... it is not necessary to add a dedicated list of the studies used in the work. Instead, the authors should include those references (i.e. those used in the systematic review) within the text, the tables etc. in such a way that the final "Reference" list would end up with >=58 references.
Author Response
Responses to Reviewer:
- Remove colon (:) from the "Results" heading.
Thank you for the suggestion as it makes the paper more uniform. The change has been reflected in the paper.
- Now, the references... it is not necessary to add a dedicated list of the studies used in the work. Instead, the authors should include those references (i.e. those used in the systematic review) within the text, the tables etc. in such a way that the final "Reference" list would end up with >=58 references.
Thank you so much for your comments. We have combined the references into one list as mentioned and incorporated all the references into the text as you have suggested. Thank you for the excellent suggestion.
- Author contributions should be listed individually.
Thank you for your suggestion. We have listed individual author contributions in the updated version.